# Develop Control Architectures to Enhance Soft Actuator Motion and Force

**Mustafa Hassan** [1,*] , **Mohammed Ibrahim Awad** [2] **and Shady A. Maged** [2]

1   Industrial Automation Department, Information Technology Institute, Faiyum 12577, Egypt
2   Mechatronics Department, Faculty of engineering, Ain Shams University, Cairo 11511, Egypt
*   Correspondence: mos.m.hassan@gmail.com

**Abstract: Study:** Soft robots can achieve the desired range of motion for finger movement to match their axis of rotation with the axis of rotation of the human hand. The iterative design has been used to achieve data that makes the movement smooth and the range of movement wider, and the validity of the design has been confirmed through practical experiments. **Limitation:** The challenges facing this research are to reach the most significant inclined angle and increase the force generated by the actuator, which is the most complicated matter while maintaining the desired control accuracy. The motion capture system verifies the actual movement of the soft pneumatic actuator (SPA). A tracking system has been developed for SPA in action by having sensors to know the position and strength of the SPA. **Results**: The novelty of this research is that it gave better control of soft robots by selecting the proportional, integral, and derivative (PID) controller. The parameters were tuned using three different methods: ZN (Ziegler Nichols Method), GA (Genetic Algorism), and PSO (Particle Swarm Optimization). The optimization techniques were used in Methods 2 and 3 in order to reach the nominal error rate (0.6) and minimum overshoot (0.1%) in the shortest time (2.5 s). **Impact:** The effect of the proposed system in this study is to provide precise control of the actuator, which helps in medical and industrial applications, the most important of which are the transfer of things from one place to another and the process of medical rehabilitation for patients with muscular dystrophy. A doctor who treats finger muscle insufficiency can monitor a patient's ability to reach a greater angle of flexion or increase strength by developing three treatment modalities to boost strength: Full Assisted Movement (FAM), Half Assisted Movement (HAM), and Resistance Movement (RM).

**Keywords:** soft manipulators; FEM; control strategy; PID tuning; SPA

## 1. Introduction

A continuum robot is a type of robot that is characterized by infinite degrees of freedom and a large joints number. These properties allow connected manipulators to modify their shape at any point along their length, allowing them to work in confined spaces and complex environments where standard hard-link robots cannot operate. The soft actuator we discuss in this paper is a type of continuum robot [1].

The human hand is a vital organ as it allows a person to fulfill numerous daily life activities, such as working, drinking, and eating. Therefore, hand disability is one of the most common diseases that negatively affects a person's psychological, social, and economic life. Amyotrophic lateral sclerosis, deficiency of main minerals such as potassium and magnesium, a herniated disc in the neck, and strokes are among the most common diseases that cause weak hands [2]. Weaknesses related to scenarios mentioned above require some rehabilitation so that the hand can recover. There are three main methods used to rehabilitate the hand: full assistive motion (FAM), half assistive motion (HAM), and resistive motion (RM) [3]. In FAM, with repeated flexion and extension along a predetermined course applied to the joint, patients remain relaxed without pain [4]. HAM works by making the patient attempt to perform movements to the best of his ability while

using a robot. The actuator helps him to complete the desired movement. It has a significant impact on increasing the patient's muscle mass [5] and the tensile strength of the tissues, as well as the stimulation of the muscle through the electrical signal. There has been a fantastic development in finger rehabilitation devices in recent years. However, its functionality is somewhat limited and ineffective in many scenarios [6,7]. There are two hand rehabilitation products that are currently in use: the end exoskeleton and the effector.

There is an end-effector sensing element interface at the end of the actuator, where movement is generated by end-effector activities [8]. Exoskeleton-based devices are instead connected along the human hand, where they conform to the anatomical structure of the finger with actuators placed on the axis of the corresponding joints [9,10]. Commercially available hand rehabilitation devices, such as in-motion hands (Bionic Labs, Watertown, MA) [11], achieve simple pleats and stretches while depending on the biomechanics of the patient's hand to know movements at the joint. By taking in consideration the movement requirements of Full Assistant Motion (FAM), Half Assistant Motion (HAM), and Resistive Motion (RM) devices with minor pleat and stretch movements are often sufficient for HAM and RM. However, FAM needs hardware that can help in many activities of daily living. Then, an automated exoskeleton that controls the individual joint, has bidirectional movement, and can move freely in three-dimensional space [12].

The main contribution of this paper is to get better control of soft robots by selecting a PID controller and tuning its parameter using three different methods: ZN (Ziegler Nichols Method), GA (Genetic Algorism), and PSO (Particle Swarm Optimization). We also relied on methods 2 and 3 for the optimization techniques in order to achieve the minor RMS error and settling time with minimum overshoot [13,14], and to provide modeling based on experimental data from sensor readings and their comparisons of actual results of experiments to soft actuators. When a data-driven modelling approach is introduced, derived experimental models will have been used to control the bending response and the generated force of soft pneumatic actuators (SPA) based on feedback. A cascaded controller can be used to find the bending angle, which is based on analytical models that specifically describe the behavior of soft pneumatic actuators [15,16]. The approach presented here is not solely dependent on the properties of a particular actuator because it relies on the experimental and empirical data generated by actual tests, implicitly demonstrating differences that are complex to model mathematically. However, the first goal is to generate experimental data demonstrating the impact various pressure inputs to SPA movement so that innovative models of this approach can be tested in different scenarios. Hence, equipping the SPA with sensor capabilities is necessary to get the adequate feedback [17,18]. This then illustrates the comparison between different tuning techniques to get a better response of SPA. The paper goes on to provide a summary of the manufacturing process [19,20] that can be followed to create a typically curved trowel with a flexible sensor built inside. The curve representing the relationship between the angle of curvature of the actuator and the motion sensor was created through practical experiments [21]. In the platform of a pneumatic control circuit, the derived empirical model is used to control the bending angle by using more than one model to achieve the lowest error rate. The results obtained using both methods are presented. Finally, the paper ends with a detailed presentation of the results and the operator's ability to hold objects [22].

## 2. Materials and Methods

A complete visualization of all the work that has been presented in the form of a framework consists of 3 stages: design, modeling, and control of the soft pneumatic control, as shown in Figure 1a, with pictures showing each stage, as shown in Figure 1b.

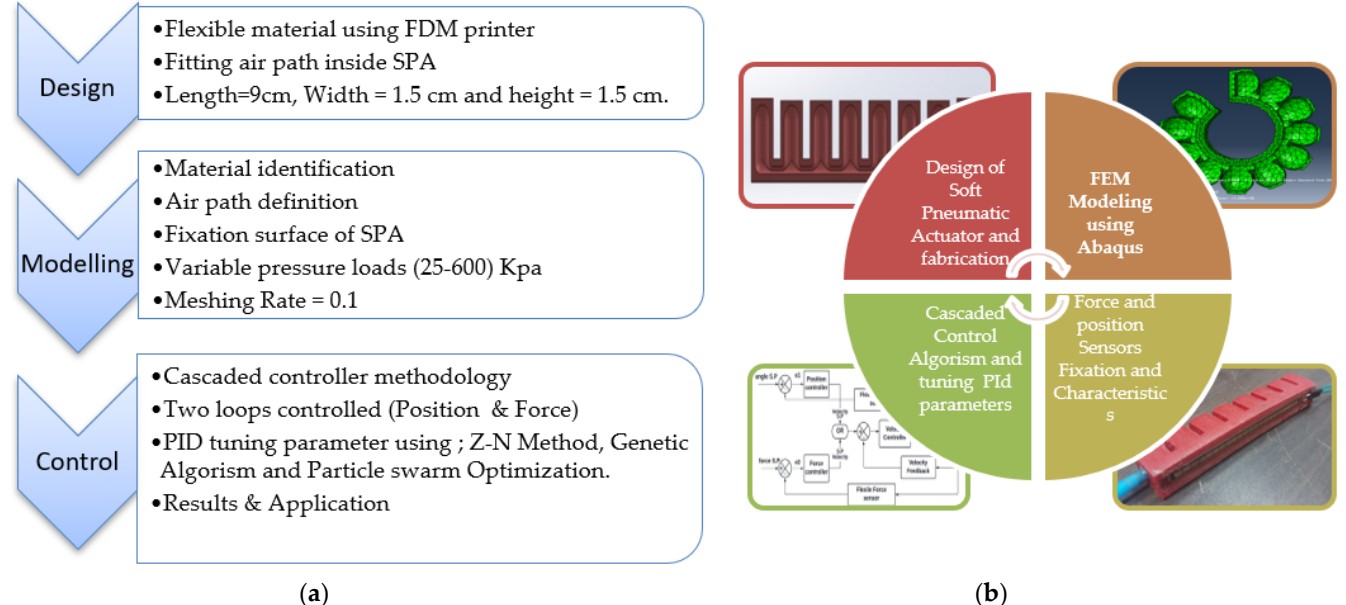

**Figure 1.** (**a**) Framework to illustrate workflow; (**b**) Graphical representation to work sequence.

As shown in Figure 2, the soft robotic glove device is divided into three main components: (1) soft pneumatic actuator (SPA), (2) fixation base, and (3) air tube.

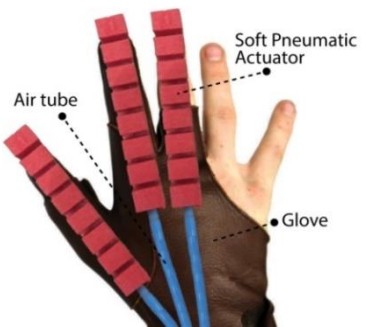

**Figure 2.** Soft pneumatic actuator (SPA).

The Assistive Hand Device (AHD) is designed to be a compact system and consists of a portable unit. The System weighs approximately 650 g without the air source, including the power supply (12 V-2 Amp), actuators (SPA), 3D printed hand support, laser cutter structure (base, cover), pneumatic tubes, and circuit board.

### 2.1. Actuator

There are many ways to manufacture SPA, most of which depend on pouring raw silicone rubber (Eco Flex-55) into a mold and mixing it with a hardener to make it hard, and making the mold out of plastic using a 3D printer stump. Another way to make SPA is by using a flexible plastic material (Leefung) using a 3D FDM printer.

The final method is the method used in this research. The actuator dimensions were dependent on the previous works that characterized the angle response and output force from a pneumatic actuator (length of 9cm, width of 1.5 cm). The manufacturing steps are as follows:

Actuator Design: The design of the finger must take into consideration the space in which the air travels, allowing an appropriate angle of curvature.

The design is printed precisely by selecting parameters (100% infill, grid infill pattern, and 0.1 mm layer height) so that the air path that leads to the curvature of the finger is not obstructed, and can locate the air path, as shown in Figure 3b.

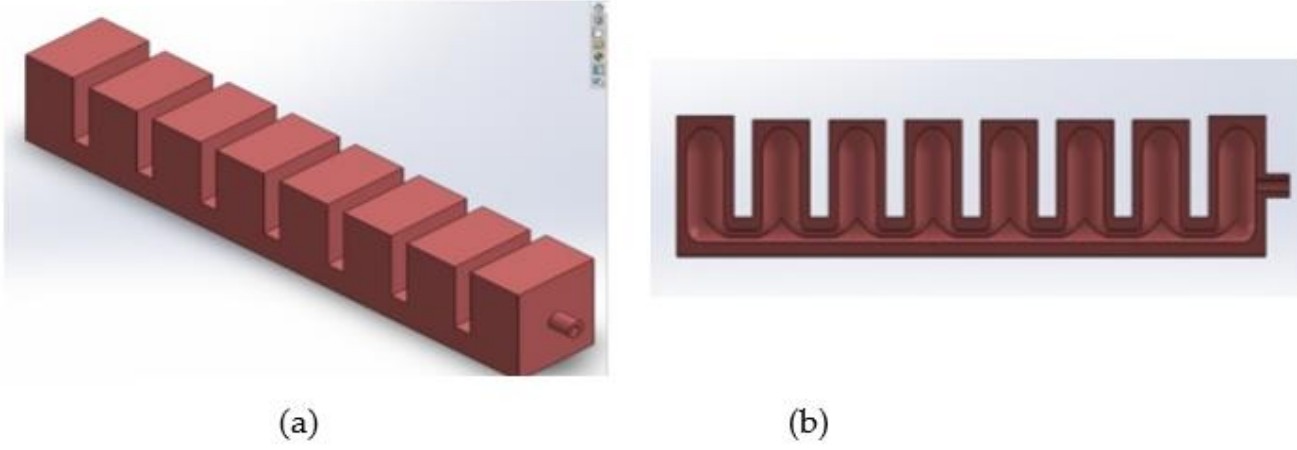

(a)            (b)

**Figure 3.** (**a**) Design of soft Pneumatic Actuator; (**b**) Air path in the SPA.

Embedding flex sensor: the flexible sensor is placed under the actuator in the form of a single, flexible, but non-extendable layer, which in turn converts the slanting motion of the finger into a control signal, as shown in Figure 4.

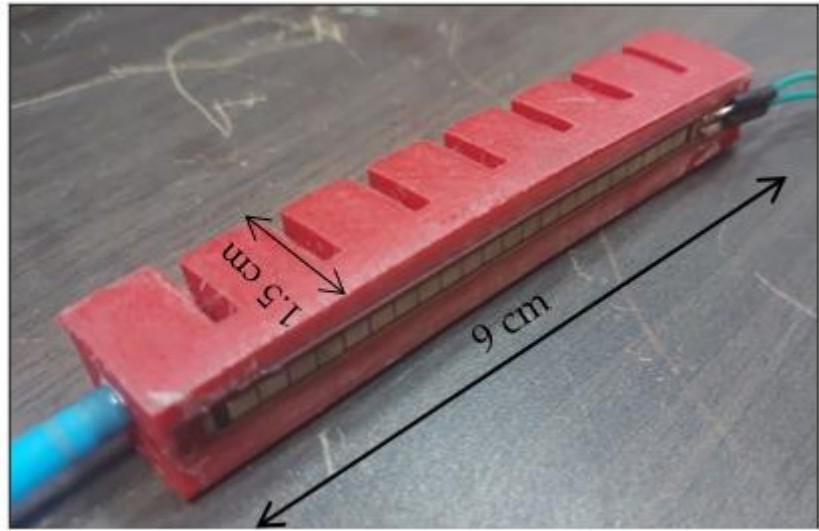

**Figure 4.** Flex sensor fixed on the soft actuator.

Actuator force sensor (FSR): the force sensor is positioned at the actuator's tip to measure the accurate force signal from the SPA to any object.

Pneumatic connection: during mechanical design, consideration is given to designing the air inlet so that the air hose is allowed to connect directly to the actuator to connect the controlled compressed air source.

### 2.2. Material Identification

The SPA model is made of flexible plastic (soft material) and manufactured using a 3D printer using FDM. This material has atypical behaviour, so it must be tested and have its properties determined by Lloyd's comprehensive testing machine, which results in the final stress-strain curve that is used with elastic material in an FEA analysis to get the actuator material behaviour. The parameters are $C_{10}$ = 1.3484 MPa, $C_{01}$ = 4.918 MPa, $C_{20}$ = 0.006527 MPa, $C_{11}$ = 0.18521 MPa, and $C_{02}$ = 0.1975 MPa.

### Modelling by Using FEM

In light of the difficulty of designing soft robots in terms of anticipating movement in any direction and the amount of force it produces, a kinetic modelling framework has

been explained, which allows for a high degree of freedom for design and modelling processes for a range of latent soft robotic components using an Abaqus software that helps us learn the nature and behaviour of these materials in terms of the bending angle at different stresses, as well as the amount of power it produces. This has been done by writing the program in a Python file, which is implemented in ABAQUS software through its programming interface. These results represent the main element in controlling soft pneumatic actuators because the results of mathematical equations have been converted by using MATLAB program, which can simulate SPA and control the position and force produced by SPA [23]. The robotics modelling process consists of several stages, starting with the design of SPA, material selection, dividing SPA into small cubes with a 10-node quadratic tetrahedron, hybrid and constant pressure mesh. The global size is one to facilitate this study as shown in Figure 5a, determining the places of adequate installation, determining pressure loads from 50 to 600 Kpa, and finally knowing the results as shown in Figure 5b. A curve fitting method approximates the deformation features through constant curvature in three-dimensional space, as shown in Figure 5c.

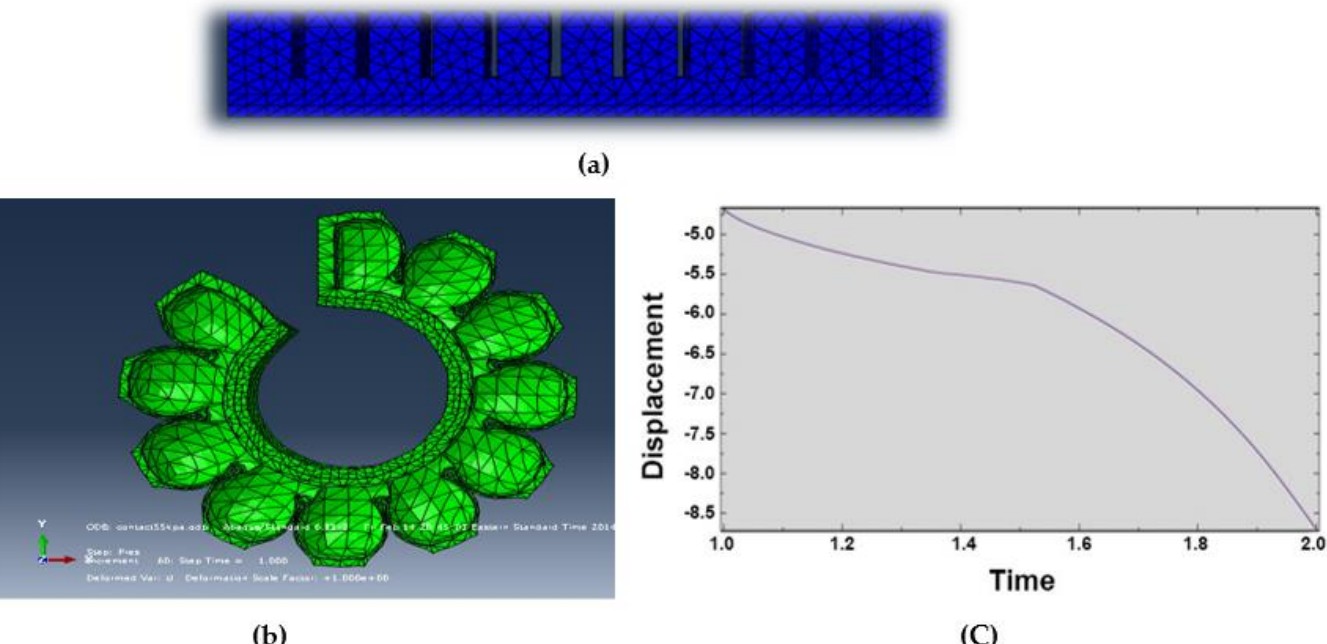

**Figure 5.** (**a**) Meshing of SPA; (**b**) Deformation of SPA; (**c**) Curvature in three-dimension space.

*2.3. Sensory Feedback*

2.3.1. Bending Sensor

The first type of sensor required is a flexible bending sensor that measure the angle of curvature of the actuator that changes with the change in internal pressure. The sensor theory works to change the sensor's resistance as a result. From the movement of the actuator, converting this change into voltage and then sending it to the microcontroller so that it can predict the bending angle of the actuator each time a different pressure is introduced, and to reach a high level of accuracy, it was necessary to test the sensor with several different pressures ranging from 25 to 600 kPa. From these practical experiments, the effect of the pressure difference on the bending angle of the actuator is shown in Figure 6. Moreover, a period has been determined in about 500 milliseconds, the pressure has gradually increased, and the trigger has been monitored by the angle achieved in this period, which then helps us in controlling when the actuator reaches the maximum curvature, providing us flexibility in the rehabilitation process.

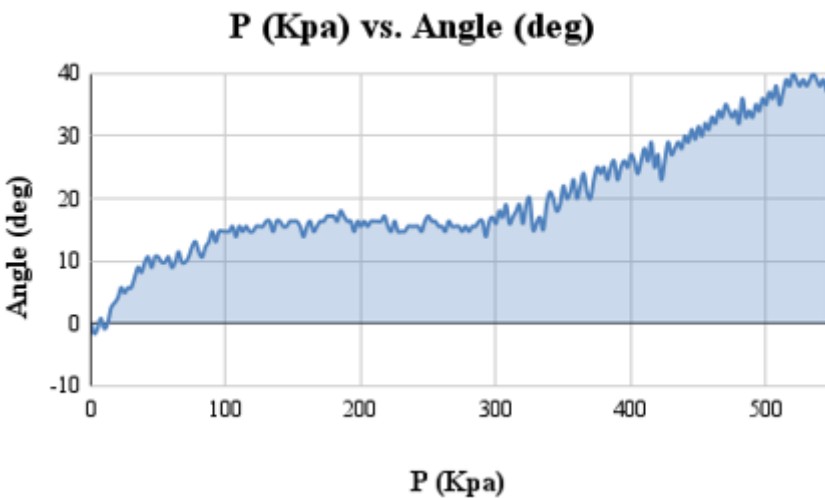

**Figure 6.** Relation between input pressure and bending sensor readings.

### 2.3.2. Bending Sensor Calibration

A fast-response camera running at 120 fps was used to capture the curvature of the actuator from its original position and the maximum number of frames captured during playback. The internal and external camera parameters have been calibrated to reduce the average calibrated error to 0.05 mm. They measured the curvature using Python and OpenCV software and split SPA into three points on the finger's first, middle, and end. When the finger moves, a triangle with known lengths is used to calculate the angle of bend concerning the axis passing through the rule of the soft operator. During a typical test, the camera is powered externally via a microcontroller so that each captured image frame has a calculated bend angle value that can be synchronized with the sensor feedback. The test platform explains the installation of the webcam and SPA, so that the calculated bending angle for each image frame captured can be compared with the corresponding sensor feedback. The test platform explains webcam installation and SPA, as shown in Figure 7.

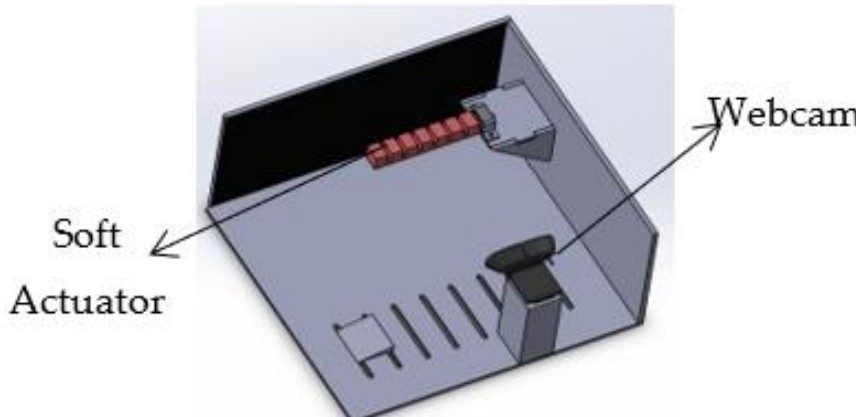

**Figure 7.** Test rig for measure bending angle values for SPA and compare it with webcam readings.

### 2.3.3. Force Sensor

The secondary feedback sensor in this proposed system is the elastic force, which changes the amount of force measured when pressing on it due to the actuator's bending motion. This force is converted into voltage and sent to the controller. This provides a measurement that can be linked to the real force from the actuator, allowing us to see how much force the actuator produces with a difference in pressure acting, as shown in Figure 8.

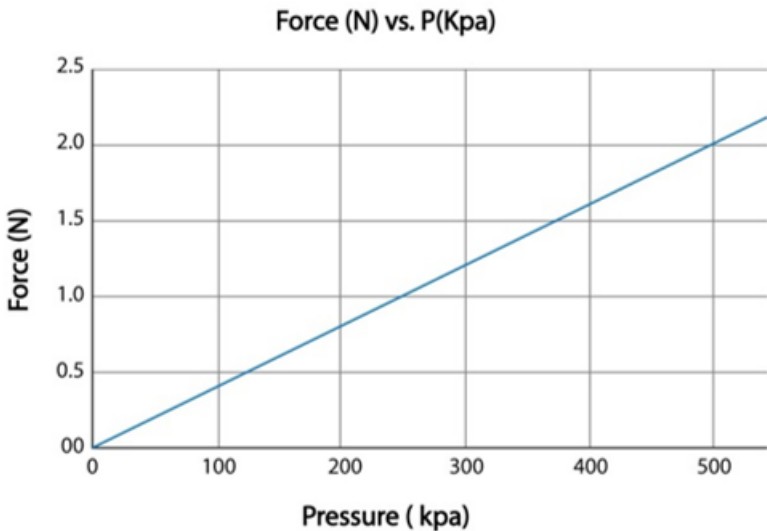

**Figure 8.** Impact of varies pressure on the output force for actuator.

### 2.3.4. Pressure Sensor

The third sensory reaction that raises interest in this work is the difference in pressure at each of the two openings of the pressure sensor (MPX10DP). Since the change in pressure has been noticed to lead to a change in angle and force resulting from SPA, it is necessary to monitor the pressure to maintain the pressure inside the system and avoid any disturbances from the source.

### 2.3.5. Pneumatic Circuit

The pneumatic system consists of an air pump (B07CGQLX2J), an air tank to maintain a constant pressure during operation, sensors to measure air pressure in the system, and quick response valves (SMC-VQ1100-U), as shown in Figure 9.

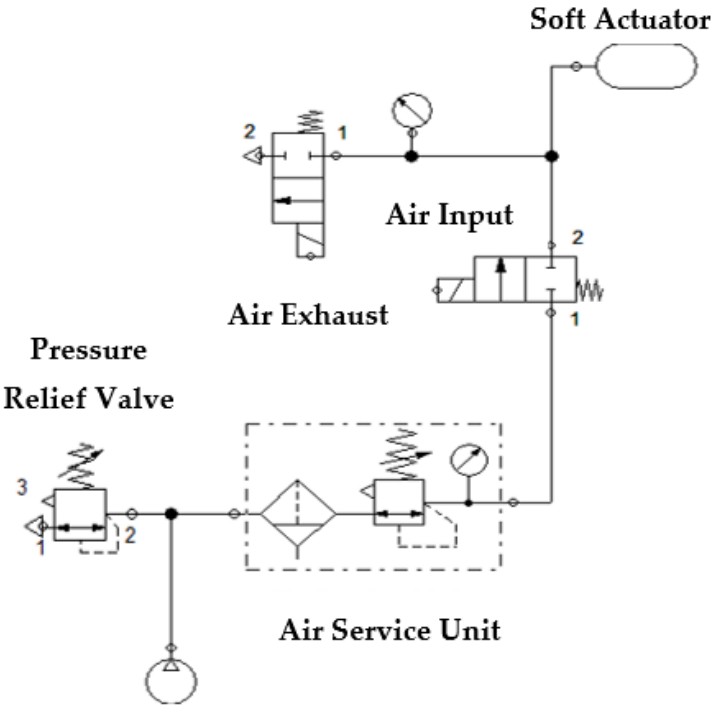

**Figure 9.** Pneumatic Circuit design for SPA.

### 3. Results

*3.1. Control Methodology*

3.1.1. Internal Pressure Control Using Fast-Response Valves

SPA contains many degrees of freedom, so most previous research avoids complex control techniques to achieve results. In this paper, the SPA was realized with hybrid technology, using two different dynamic models in the same system. A fair amount of research has been done on modelling hybrid systems, but it was only recently that the analysis and synthesis of hybrid control laws have begun to appear in the literature. One of the main objectives of this paper is to broaden the range of knowledge concerning hybrid systems. To this end, tools have been developed to expedite the analysis and the design of mixed control laws by changing the pressure in the air entering the actuator via control valve. (SMC-VQ110U), which is operated via a pulse width modulation (PWM) signal at a frequency of 40 Hz. Pressure and force sensors are connected to calculate the bending angle, force generated, and send the signal to the microcontroller.

3.1.2. Cascaded Control Loop

The three operating modes are FAM, HAM, and RAM. Each mode can be controlled in two different ways: controllingl the position of the finger and controlling the force generated by the finger. To accurately control the positioning and force control, a block diagram has been represented for a cascaded control loop to control the soft actuator in two different circuits, one to control the bending angle and the other to control force, as shown in Figure 10. In positioning control, controlling the speed of the finger is a must (inner loop) by controlling the amount of air inside the finger to avoid any deformation. The set-point (S.P) of the finger velocity varies according to angle S.P or force S.P (outer loop). Moreover, there was a difference between the design, its accuracy, striving to reach a higher rate of bending angle, the ability of the material to withstand greater air pressure, and the method of controlling this robot, especially since it is n elastic material, hence the importance of using the successive control ring for the angle and others in order to be able to accurately control the soft robots.

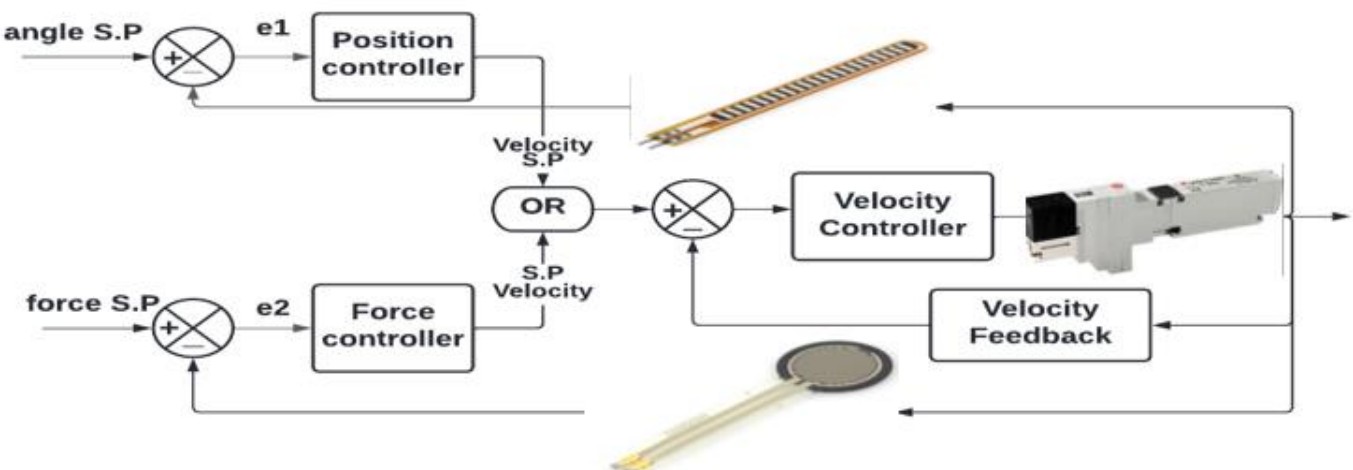

**Figure 10.** Block diagram to control SPA.

3.1.3. Switching Control

Switching control is implemented by a change between two modes of operation by using state flow in Simulink. The first state is velocity control, and the second is force control. The transition from velocity control to force control occurs at velocity equals zero.

However, the transition from force control to speed control occurs at force = 0. This depends on controlling the velocity before touching the object, and then controlling the force during the hold.

### 3.1.4. PID Tuning Parameters

Control units have been used in many applications, such as speed, power, and position. One of the most commonly used methods is PID, which helps us reach a steady state and error = 0 in the least possible time. Factors have been determined through several methods, the most famous of which is the ZN. In many applications, this method is sufficient to achieve the missing parameters of the controller. However, with some more challenging applications and somewhat complex control systems, new ways have emerged for finding PID parameters using optimization techniques, such as genetic algorithms (GA) and particle swarm optimization (PSO). This method allows us to achieve better results and compare tuning parameters for different techniques in Table 1.

**Table 1.** PID Optimized Parameters.

| Tuning Method | KP Inner Loop | KI Inner Loop | KP Outer Loop | KI Outer Loop |
|:---:|:---:|:---:|:---:|:---:|
| Z-N PID | 0.9 | 1.5 | 2 | 60 |
| GA-PID | 0.6 | 1.2 | 1.723 | 32 |
| PSO-PID | 0.65 | 1.38 | 1.6 | 29 |

1.  Tuning PID parameter by using Z-N Method;
    The first method used to find the parameters of the PID controller is the ZN tuning method, which is a closed-loop control method based on determining the maximum system gain period for stability. To minimize errors, the integral coefficient has been provided gradually. To avoid fluctuation, the differential coefficient has been given a value; all initial values are in tables from which one can extract the coefficients' initial values.

2.  Tuning parameter by using Genetic Algorism (GA);
    A genetic algorithm is one of the new methods used to determine the parameters of PID. This process can be done by forcing a set of numbers for the parameters and comparing them to change the error rate if it improves or worsens. Through this process, a colossal group of numbers can be tested until they get the best three numbers for them and the lowest value of the average error that occurs. It helps the controller have a faster and more stable response time.

3.  Tuning parameter by using Particle Swarm Optimization (PSO);
    In 1995, Kennedy and Eberhart introduced the particle crowd optimization (PSO) method. It is an optimization technique and a kind of evolutionary computation technique [24,25].
    In this method, the initial values have been assumed of the microcontroller user's PID coefficients, and for each number, evaluation fitness can be calculated and compared to those with the other values to reach the least mean square error possible until the values have been found. These values are called P-best. When using the GA-based PID controllers, the performance also improves slightly. Different performance indices give different results. These are shown in Table 2. Comparison between the control outputs of the cascaded controller using three different methods for tuning the parameters of PID: 1-ZN Method, 2-Generic algorism, and 3-Particle swarm optimization, as shown in Figure 11.

**Table 2.** Step Response Performance for PID Controller.

| Tuning Method | Overshoot (%) | Settling Time (s) | RMS Error(%) |
|:---:|:---:|:---:|:---:|
| Z-N PID | 29 | 5.2 | 0.7083 |
| GA-PID | 3.8 | 2.6 | 0.6840 |
| PSO-PID | 0.1 | 3.1 | 0.8352 |

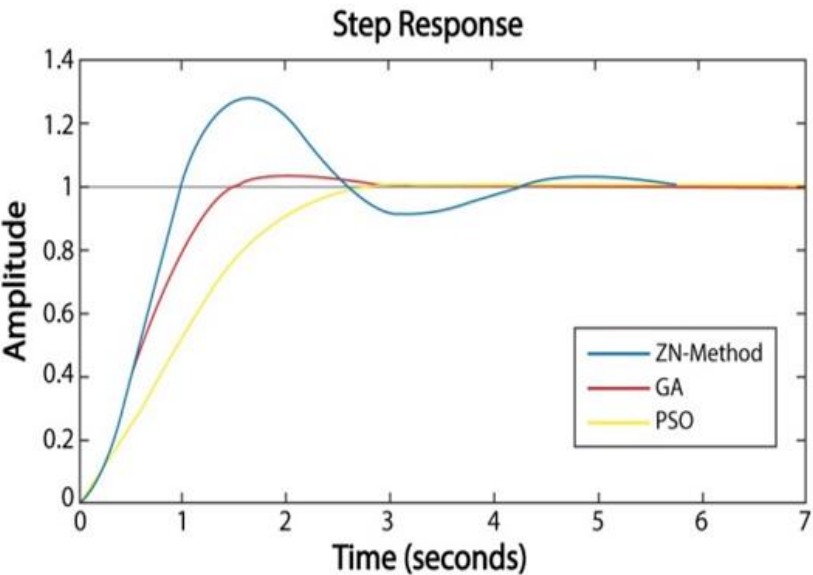

**Figure 11.** Comparison of the modelling step response for PID controller.

### *3.2. Critical Analysis and Discussion*

This study explains the cascading control technique with PID controllers and the difference between tuning parameters by using both controllers' ZN, GA, and PSO methods. It also shows the difference between system modelling and experimental positioning control in three cases, as shown in Figure 12.

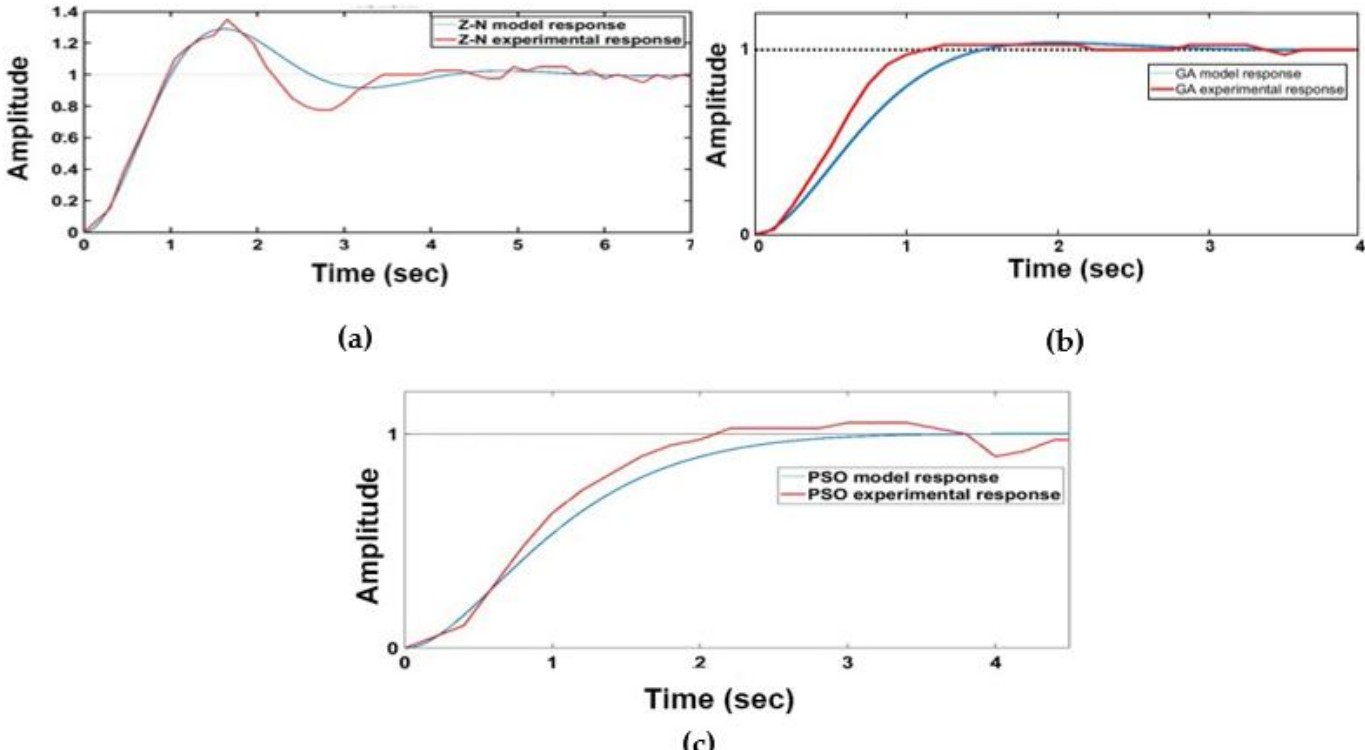

**Figure 12.** (**a**) Model & Experimental results by using Z-N method; (**b**) Model & Experimental results by using GA method; (**c**) Model & Experimental results by using PSO method.

Figure 13 shows the effect of changing the input signal (setting point) of the controller on each force generated by the actuator (output signal), the pressure inside the actuator, and the error rate between the input signal and the output signal, which indicates the

efficiency of the controller. Figure 14 illustrates the same experiment as in Figure 13, but this time with a different controller output signal. In this case, the actuator's bending angle is the output signal.

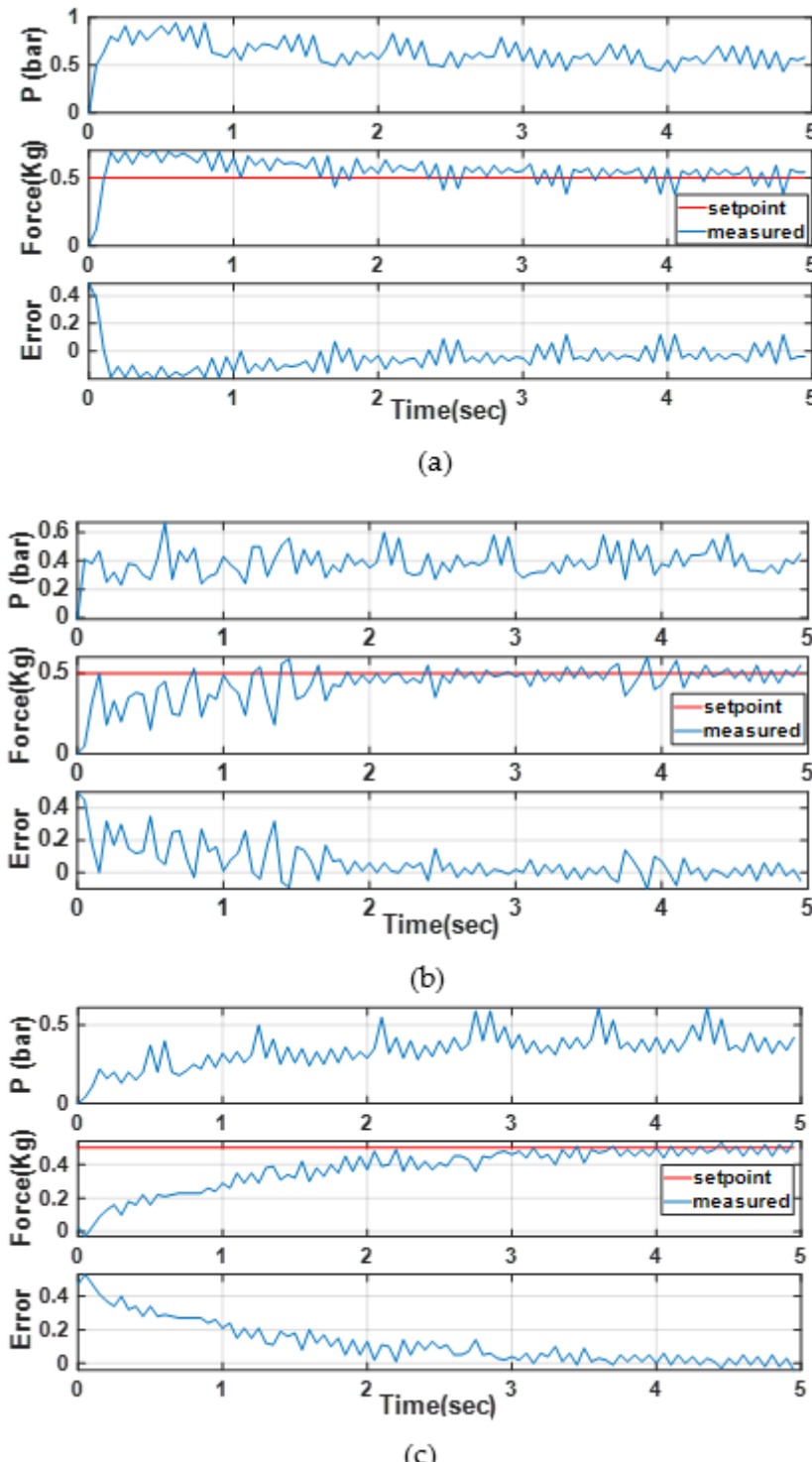

**Figure 13.** Experimental results of the change of pressure, force, and error with time by using: (**a**) Z.N tuning technique; (**b**) GA tuning technique; (**c**) PSO tuning technique.

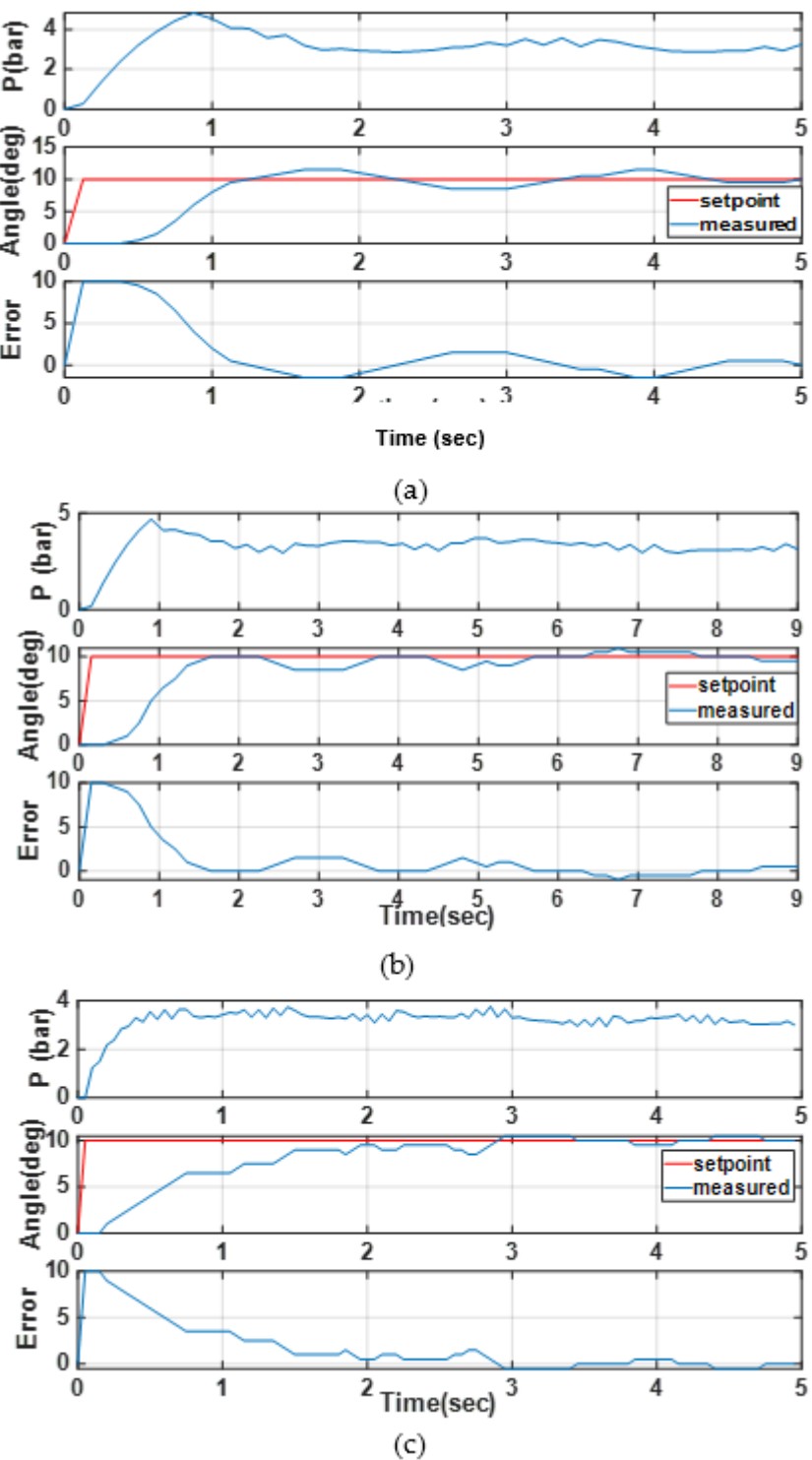

**Figure 14.** Experimental results of the change of pressure, angle and error with time by using: (**a**) Z.N tuning technique; (**b**) GA tuning technique; (**c**) PSO tuning technique.

FEA simulations prove that the force produced by SPA varies with the inlet pressure. To check this behavior, two grippers were tested on the same stand. The grippers have been tested under the pressure (4: 5 bar scale/0.5: 0.6 absolute MPa) and the holder geometry with a distance of 7 cm between two actuator holders. The test shows that grippers were tested for carrying various loads and surfaces (metal, plastic, wood, rough plastic), weights (50, 100, 200, 250 g), and complex engineering shapes (mechanical tools, bottles of water, medicine boxes as shown in Figure 15.

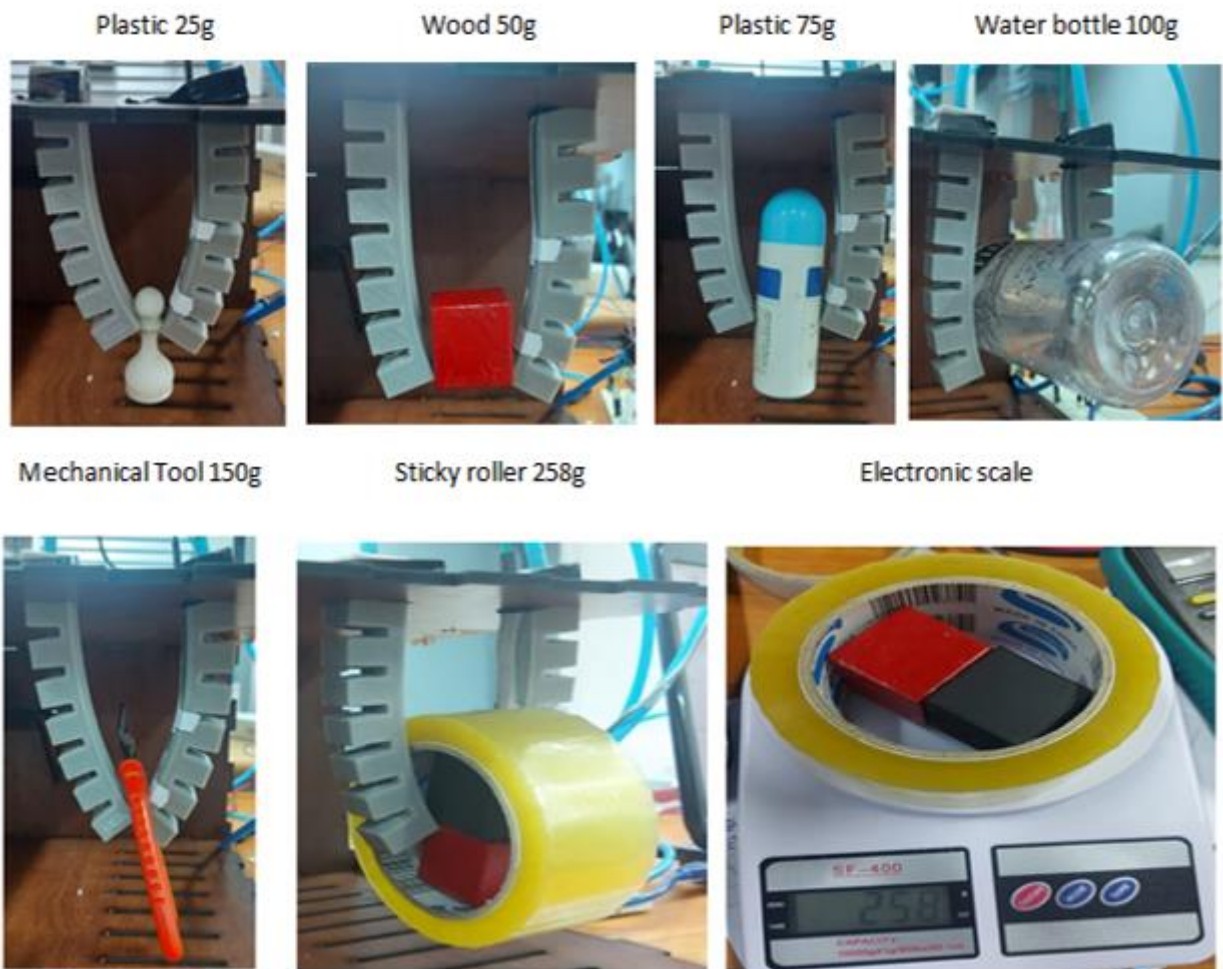

**Figure 15.** Different loads and different material.

Many research papers dealt with the issue of controlling the soft actuator in several ways, for example, using the traditional PID controller, where the RMS error rate reaches 0.92% with a long settling time of 5.2 s [26], and the feed-forward controller, which helps to improve the speed of decision-making, leads to a reduction of the settling time to 3 s [27], minimizes RMS error to 0.7%, and is also a cascading control system, in which the set-point of the control unit is determined by another control unit. This is the proposed system in this study, with the addition of tuning the PID parameter through optimization techniques as genetic algorism and particle swarm optimization, which gets the lower values of RMS error of 0.6% and 0.83%, respectively, and settling time 2.5 s and 3.1 s, respectively. This is the leading advantage in our proposed system to achieve precise control of the position of the actuator and maintain accuracy when increasing the force coming out of it.

The strength of this research lies in providing accurate control of soft robots by using Cascade Controllers and tuning PID Parameters by using Optimization Techniques to reach the minimum error rate, as shown by graphs 13 and 14. One of the most important challenges facing this proposed system is to reach the most significant angle with an increase in the force generated by the SPA, which is the most complicated matter, while maintaining the desired control accuracy. The impact of this research in real life helps in many medical and industrial applications, the most important of which is the transfer of things from one place to another and the process of medical rehabilitation for patients with muscular dystrophy.

## 4. Conclusions

In this proposed system, the actuator has been designed and implemented through a pneumatic circuit, monitoring pressure [28], force, and bending angle of SPA in order to reach a precise level of control of the position and the force [29] coming out of it by using a PID controller. Its parameters are tuned in different ways of optimization, such as Genetic Algorism and Particle Swarm Optimization, to reach the suitable performance and compare them with the traditional methods of tuning parameters, as is the ZN method [30]. It turns out that PSO is the best way to avoid overshooting with 0.1, but GA reaches the settling time in the least possible time of 2.6 s, and it has the lowest value of the error with 0.6840. This is evident from the previous curves. It also reaches a loading weight of 300 g while maintaining precise control of the SPA [31].

In the future, the soft robot will seek to test under different operating conditions to reach the largest angle of curvature and the most significant weight that the robot can carry for use in medical and industrial applications while maintaining precise control.

**Author Contributions:** S.A.M., M.H. and M.I.A. declare they contributed equally for this manuscript. All authors have read and agreed to the published version of the manuscript.

**Funding:** Funded by [Moustafa Hassan].

**Institutional Review Board Statement:** Not applicable for studies not involving humans or animals.

**Informed Consent Statement:** Informed consent was obtained from all subjects involved in the study.

**Conflicts of Interest:** The authors declare no conflict of interest.

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
