# Peer review of "Develop Control Architectures to Enhance Soft Actuator Motion and Force"

_computation, doi:10.3390/computation10100178_

Round 1

Reviewer 1 Report (Previous Reviewer 1)

I read the updated version of the paper and found that all my concerns have been improved. For my part, I have no comments.

Author Response

Than you for your feedback .

Reviewer 2 Report (Previous Reviewer 2)

1)    The figure number and captions are missing for the first figure under section 2. Hence all the figure numbers and cited in text should be changed accordingly.

2)    A general high level block diagram or framework of technical work with description in this paper is required to add at the beginning of chapter 2. Although you have tried to present a figure which is not sufficient. You need to modify the figures with specific details.

3)    Very limited discussion and analysis is given for the figures 12 and 13. Please expand them with critical analysis.

4)    Further experimentation with results is expected.

5)    Performance/advantages comparison with existing related works has not been given at the end of result section to validate the capability of the proposed method presented in the paper.

6)    Please add strength, limitation and impact/significance of this work in real life scenarios under “critical analysis and discussion” section.

Author Response

My response to the comments is in the attached file name: Details response for reviewer no.2.

Reviewer 3 Report (New Reviewer)

doc file attached.

Author Response

My response to the comments is in the attached file name details response for reviewer no.3.

Round 2

Reviewer 2 Report (Previous Reviewer 2)

Please add the responses of my previous following comments in the revised manuscript. 

5)    Performance/advantages comparison with existing related works has not been given at the end of result section to validate the capability of the proposed method presented in the paper.

6)    Please add strength, limitation and impact/significance of this work in real life scenarios under “critical analysis and discussion” section.

Author Response

The response to comments in the attached file as round2-Details response for reviewer no.2.pdf

This manuscript is a resubmission of an earlier submission. The following is a list of the peer review reports and author responses from that submission.

Round 1

Reviewer 1 Report

1.       The advantages/merits of the robotic design are unclear. Explicitly summarize the contribution and significance of the paper, which is not quite clear in the current form (the design? The model? Or else?)

2.       Provide video demonstration of the developed robot

3.       Use high-resolution images for figures.

4.       Many latest pneumatics/hydraulics-actuated robots and some follow-the-leader continuum robots are missing from the literature.

5.       I strongly recommend that the authors conduct further study on this robot.

Author Response

Develop Control Architectures to Enhance Soft Actuator Motion and Force

Response to Reviewer No.1 Comments

We would like to thank the reviewer for their constructive comments, which have considerably improved the quality of the paper.

  1. The advantages/merits of the robotic design are unclear. Explicitly summarize the contribution and significance of the paper, which is not quite clear in the current form (the design? The model? Or else?)

The advantages of this research are that it gave better control of soft robots by selecting a PID controller and it was tuning parameters by using three different methods: ZN (Ziegler Nichols Method), GA (Genetic Algorism) and PSO (Particle Swarm Optimization). We also relied in Method 2 and 3 on the optimization techniques in order to reach the least RMS error, settling time and minimum over shoot as shown in Table No. 2

  1. Provide video demonstration of the developed robot

https://drive.google.com/drive/folders/1SnspFiFIZqONfW9iqtdE3PLbx3NICC5Q?usp=sharing

  1. Use high-resolution images for figures.

Yes, we reviewed all graphs

  1. Many latest pneumatics/hydraulics-actuated robots and some follow-the-leader continuum robots are missing from the literature.

Thank you for your suggestion. The paper has been revised, proofread, and adjusted based on the reviewer suggestions. You find this in the first part in introduction.

  1. I strongly recommend that the authors conduct further study on this robot.

Yes, we are already working on this robot and subject it to different operating conditions, some of which were mentioned in this research, which is the ability to carry different weights and sizes up to 300 grams, and we strive to reach an angle of curvature and the ability to carry larger weights and you can find it in the application part in figure 14.

Reviewer 2 Report

1)    Numeric achievement is missing of your work both in Abstract and conclusion.

2)    The novelty of this paper is not very clear. Would you please add the novelty of the work either in abstract or in conclusion which differs from literature?

3)    Please review more relevant works and find the research gap from there.

4)    A general high level block diagram or framework of complete technical work in this paper is required to add at the beginning of chapter 2 so that readers can follow up the entire technical work has been done in this paper.

5)    The entire experimental set up with description is missing. Please add and describe properly.

6)    Figure 3 should be labelled properly and fig 4b is not visible. Please change.

7)    Description and analysis should be enhanced for the Figures 9, then 10-14.

8)    Please also discuss Table 1 and Table 2.

9)    Result section is too small. Further experimentation with results is expected.

10) Performance/advantages comparison with existing related works should be added at the end of result section to validate the capability of the proposed method presented in the paper. Results section should be updated by adding a subsection as “critical analysis and discussion”.

11) Please add strength, limitation and impact/significance of this work in real life scenarios.

12) Specific Future research directions are missing. Please add those at the end of conclusions.

Author Response

Develop Control Architectures to Enhance Soft Actuator Motion and Force

Response to Reviewer No.2 Comments

We would like to thank the reviewers for their constructive comments, which have considerably improved the quality of the paper.

  • Numeric achievement is missing of your work both in Abstract and conclusion.

This note was taken into account and the introduction and the conclusion were reviewed, clarifying the numerical numbers that illustrate the results of this research.

  • The novelty of this paper is not very clear. Would you please add the novelty of the work either in abstract or in conclusion which differs from literature?

The novelty of this research are that it gave better control of soft robots by selecting a PID controller and it was tuning parameter by using three different methods: ZN (Ziegler Nichols Method),    GA (Genetic Algorism) and PSO (Particle Swarm Optimization). We also relied in Method 2 and 3 on the optimization techniques in order to reach the least RMS error, settling time and minimum over shoot as shown in the final part of abstract and Table No. 2.

  • Please review more relevant works and find the research gap from there.

Yes. Previous research related to soft robotics was reviewed and we found that the challenge lies in finding a method for controlling soft robots that is accurate and can be used in both medical and industrial applications, which is what we tried to cover in this research. We find it at the end of cascaded control loop part.

  • A general high level block diagram or framework of complete technical work in this paper is required to add at the beginning of chapter 2 so that readers can follow up the entire technical work has been done in this paper.

Thank you for your suggestion. The paper has been revised, proofread, and adjusted based on the reviewer suggestions.

  • The entire experimental set up with description is missing. Please add and describe properly.

The soft robot is installed as in Figure 7 so that the webcam is perpendicular to it, and then we can calibrate the amount of inclination angle that results when affecting it with variable pressures, and there is another part in our test rig, which is shown in Figure 15 which is the clamping of two of the Soft actuators face each other to know the ability to carry things and calibrate them by using a digital scale.

  • Figure 3 should be labelled properly and fig 4b is not visible. Please change.

Thank you for your suggestion. The paper has been revised, proofread, and adjusted based on the reviewer suggestions.

  • Description and analysis should be enhanced for the Figures 9, then 10-14.

Figure 9: Block diagram for cascaded control loop to control soft actuator in two different circuits one them to control bending angle another to control force

Figure 10: Comparison between the control outputs of the Cascaded Controller using three different methods for tuning the parameters of PID: 1- ZN Method, 2- Genetic algorism, 3-Particle swarm optimization.

Figure 11: comparison between Modelling and Experimental results by using (a) Z-N method, (b) GA method, (c) PSO method.

Figure 12: A practical comparison between the changes of the internal pressure inside the soft actuator and the extent of its impact on its bending angle and the error rate over time.

Figure 13: A practical comparison between the changes of the internal pressure inside the soft actuator and the extent of its impact on its force and the error rate over time.

Figure 14: A practical test of the soft actuator, which carries weights of up to 300 grams and carries different shapes in terms of size and material.

  • Please also discuss Table 1 and Table 2.

We used the cascade controller, which is divided into two primary circuits expressing angle control, and the second expressing force control. For each inner and outer loop, the first table shows the tuning parameters (kp, Ki) for both the inner and outer loops of each circuit and for each method the tuning parameters.

While the second table indicates the result of applying different tuning methods such as ZN Method, Genetic Algorism, and Particle Swarm Optimization to the root main square error (RMS), settling time and over-shoot.

  • Result section is too small. Further experimentation with results is expected.

Divide the results into three sections, the first of which is a description of the type of control used and how to use it, which is the Cascade Control Loop, which contains two loops, one of which controls the angle and the other controls the force, and the transition between them is through the speed sensor, and this is called switch control. As shown in graph No. 9.

And in the second part of the results section, we talked about the methods used to tuning the controller parameters. This is evident in Table No. 1 and the differences between them in the model in graph No. 10 and found in real No. 11.

While the third part shows the effect of different methods in tuning parameters on the accuracy of the controller, and this is available in graphs 12 and 13.

  • Performance/advantages comparison with existing related works should be added at the end of result section to validate the capability of the proposed method presented in the paper. Results section should be updated by adding a subsection as “critical analysis and discussion”.

Thank you for your suggestion. The paper has been revised, proofread, and adjusted based on the reviewer suggestions.

  • Please add strength, limitation and impact/significance of this work in real life scenarios.

Strength: The strength of this research lies in providing accurate control of soft robots by using Cascade Controller and tuning PID Parameter by using Optimization Techniques to reach the minimum error rate, as shown by graph No. 12 and 13.

 Limitation: The challenges facing this research are to reach the largest bending angle and then increase the force generated by the finger, which is the most complicated matter while maintaining the desired control accuracy.

 Impact: The effect of this research paper is to provide precise control of the robot, which helps in many medical and industrial applications, the most important of which is the transfer of things from one place to another and the process of medical rehabilitation for patients with muscular dystrophy

12) Specific Future research directions are missing. Please add those at the end of conclusions.

Based on the reviewer’s suggestions, In the future, we seek to test the soft robot actuator under different operating conditions to reach the largest angle of curvature and the largest weight that the robot can carry for use in medical and industrial applications, while maintaining precise control.

Reviewer 3 Report

According to the abstract, this paper presents a specific design of soft pneumatic actuator along with various control techniques. Though the idea has a good merit, the current version of the paper was very confusing confusion to me. I suggest the authors to review the main structure of the paper to clarify that:

- Is the design evoked a contribution to science or just a replication of what was done before (though required for modelling)?

- If one of the goal is truly the modelling portion (or should we talk about identification considering it is based on experimental data), then the experimental set-up, tuning and validation should be presented as such. In its current form, it appears as the methods was designed to test the sensors used to collect data...

- In the end, it is unclear to me which sensors will be required to control the SPA vs which sensors are used to acquire data in order to design and validate the model... (also note replication of sub-sections titles like 2.3.1 and 2.3.2 both entitled "bending sensor")

- Thoughout the paper, I was waiting for the model validation... I am confused that it did not come while the main objective talks about modelling. I think I understand that this step was "by-passed" and considered as done through the direct looping with the actual set of sensors. Yet, I can't help but wonder about limits analysis which are of capital importance in the presented scenario for security.

Overall, I believe the study appears interesting but that the manuscript overall organization should be revised to improve general readability and clarify the purpose of the study.

Author Response

Develop Control Architectures to Enhance Soft Actuator Motion and Force

Response to Reviewer No.3 Comments

We would like to thank the reviewers for their constructive comments, which have considerably improved the quality of the paper.

According to the abstract, this paper presents a specific design of soft pneumatic actuator along with various control techniques. Though the idea has a good merit, the current version of the paper was very confusing confusion to me. I suggest the authors to review the main structure of the paper to clarify that:

  1. Is the design evoked a contribution to science or just a replication of what was done before (though required for modelling)?

The importance of the design in this research comes because we were able to reach a greater angle of curvature by fitting the internal the air path, which in turn contributed to the entry of a larger amount of air and thus obtaining a greater bending angle as shown in figure 2.b, and then we moved to the modelling part and then the part The main thing in this research is the precise control of the soft actuator in terms of bending angle and the resulting force.

Strength: The strength of this research lies in providing accurate control of soft robots by using Cascade Controller and tuning PID Parameter by using Optimization Techniques to reach the minimum error rate, as shown by graph No. 12 and 13.

 Limitation: The challenges facing this research are to reach the largest bending angle and then increase the force generated by the finger, which is the most complicated matter while maintaining the desired control accuracy.

 Impact: The effect of this research paper is to provide precise control of the robot, which helps in many medical and industrial applications, the most important of which is the transfer of things from one place to another and the process of medical rehabilitation for patients with muscular dystrophy

  1. If one of the goal is truly the modelling portion (or should we talk about identification considering it is based on experimental data), then the experimental set-up, tuning and validation should be presented as such. In its current form, it appears as the methods were designed to test the sensors used to collect data...

As mentioned previously, the modelling is a means of knowing the expected angle of the soft actuator, and it was done by defining the material and its characteristics, the fixation surface of the robot, the internal path of the air, the air pressure inside it, the meshing, finally get shape of it after deformation as shown in Figure 4 and then We can compare it with the real angles that happen to the robot, which we can know in the part of the sensors installed on it (pressure, force and bending sensor), and this is very necessary for us to be able to control it well, and this is what we did in Cascaded Control Loops

  1. In the end, it is unclear to me which sensors will be required to control the SPA vs which sensors are used to acquire data in order to design and validate the model... (also note replication of sub-sections titles like 2.3.1 and 2.3.2 both entitled "bending sensor")

Thank you for your suggestion. The paper has been revised, proofread, and adjusted based on the reviewer suggestions.

The sensors required to validate the design are the angle sensors that show the deformation that actually occurred, and the pressure sensor, which in turn shows the relationship between the bending angle and the pressure inside the robot, while the sensors that are used to control the soft robot are:

1- Force sensor, which helps us to know the force generated by the finger at different pressures

2- Angle sensor, which is also used in the design and helps us to know the true angle of the soft actuator, and we can also differentiate this values to get the speed of the robot movement.

  1. Throughout the paper, I was waiting for the model validation... I am confused that it did not come while the main objective talks about modelling. I think I understand that this step was "by-passed" and considered as done through the direct looping with the actual set of sensors. Yet, I can't help but wonder about limits analysis which are of capital importance in the presented scenario for security.

The modelling part in this research is one of the important parts that help in achieving better control of the soft robot, and this is the main objective of the research, and I can mention the limitations of the modelling that were used, which are

1- The fixation surface of the soft robot, which is the surface connected to the air connection as shown in figure 3.

2- The air path inside the finger which is determined by the design

3- The value of pressures inside the airway from 25 to 600 kPa

4- Meshing by mean divide the actuator into very small portions (10-node quadratic tetrahedron)

I added a framework at the first of the paper to clarify the work in this paper.

Overall, I believe the study appears interesting but that the manuscript overall organization should be revised to improve general readability and clarify the purpose of the study.

Thank you for your advice.

Round 2

Reviewer 1 Report

I read the revised paper and all my concerns have been modified well. For my part, I accept this paper if some minor modifications on language could be made. 

Reviewer 2 Report

This reviewer would like thank the authors for modifying the paper according to the comment. However, following comments still needs to be addressed. 

  • Performance/advantages comparison with existing related works should be added at the end of result section to validate the capability of the proposed method presented in the paper. Results section should be updated by adding a subsection as “critical analysis and discussion”.

Reviewer 3 Report

Thank you for this revised version. However, I believe more work is still required prior to publication. Here are some comments:

Abstract

-       Abstract is missing the context of the study.

-       Please do not use abbreviation without definition…  E.g. SPA in the abstract.

-       Please revise English… for example:

o   Iterative design was used to access the data that makes the movement smooth AND the range of movement was made wider, AND the validity … 

o   Can iterative design truly be used to access data?

-       In the methods part, the authors talk about the doctor? I believe this was meant to mention that three different modes were developed… but this has to be re-worded and put in the right section describing the device/algorithm.

-       Results sub-section:  It appears as the first sentence is not a result of this study but part of the context. Second sentence appears to be the motivation for this study. Then, the PID with different tuning parameters is the method. Only the last sentence is a result. 

-       Impact: considering the paper demonstrates control theory over a very specific robot, can you really confirm that these specific results provide better control for medical and industrial applications? It has not been tested in these conditions and I believe the robot itself would be very different. It may have the potential to enhance control applications in these industries, but in its current status, I don’t think there is any proof of this…

Introduction: *Plagiarism found with grammarly

-       First paragraph:

o   First sentence: “a number of joints”: do the authors mean a large number of joints?

o   Second sentence: “to modify and modify”?

o   Nice addition to put the paper in perspective…

-       Second paragraph:

o   “human hand is the main engine”: I suggest “is a very important engine”… I agree this is very important but so are the legs to enable a person to move around… Also, I suggest reviewing the sentence for something like: the human hand is a very important engine as it allows a person to fulfill numerous activities of daily living such as eating, drinking, and working. 

o   Second &third sentences: please add a reference.

o   3 types of rehab: careful… rehab exists without robots…

o   Line 10: why is patients with a capital letter?

-       Paragraph 3:

o   Line 5: Sentence starting with “and”?

o   First sentence on page 3: “How the free bending angle…” please revise the sentence as starting the sentence with how announces a question…

o   Type of reference changes for ref 16?

o   “Then, they illustrate”: who’s they?

o   “Then, the platform, this includes…”: to be revised… do not understand the sentence.

o   English: then, then, then, then

Materials and methods

-       Framework is very useful… nice addition. Wording could be improved. I suggest using the exact same structure in the Materials & Methods section to fit the announced framework. (i.e. Section 2.1 SPA Design, Section 2.2 SPA model…)

-       Beginning of page 4. Please review the text. Suggestion: The Compact Hand-Assistive Device (CHAD? I suppose since the authors chose to put capital letters in the name) is a portable unit composed of a 3D printed hand support, a laser cut structure, pneumatic tubes, and soft pneumatic actuators (SPA), all connected to a circuit board requiring a 12V power supply and mounted on a glove.

-       Actuators: 

o   I wonder if the description of the different ways to manufacture SPA really goes in this section? This section should describe what the authors have done…

o   “The design is printed precisely”: what do the authors mean? What specific action did you do to ensure this fact? “precisely” is subjective…

-       Figure 3: I suggest adding a dime or something else on the picture to give a sense of sizing.

-       Section 2.2:

o   The SPA model is made of silicon…: remove model as I don’t believe you want to refer to your modelling steps but to the “device” itself.

o   Again, part of this section appears as justification and not methods… Right place?

o   How was the testing performed? The authors refer to Lloyd’s comprehensive testing machine. Yet, it would be nice to have either a figure and/or a description of this process. Otherwise is it worth a section? Results shown here?

o   Modelling using FEM:

§  A kinetic modelling framework has been proposed: by whom? Either please add a reference or revise the sentence if this is a specific contribution of this study.

§  First sentence is to be revised (too long and finishes with a coma?)

§  “This can be done” or this has been done?

§  Please revise English. Example: “through which can simulate”???

-       Section 2.3:

o   If I understand correctly, this section describes the embedded sensors required for the robotic assistive device to work (sensory feedback). If so, please introduce the section as such.

o   Bending sensor: “the first type of sensor that interests us” – you mean the first type of sensor REQUIRED?

o   Bending sensor calibration: “Recording 120fps photo frames with a high-speed camera is sufficient to measure the curvature of the SPA…”. Reference? Sufficient is subjective. At this point, authors describe what was done and so I suggest re-wording to state those facts. E.g. A high-speed camera running at 120fps was used to capture the curvature…

-       Again I am a bit confused at this point… From figure 9, I deduce that you will send simulated command to the device itself but how do you measure accuracy? Overshoots is on velocity not position? In the abstract, you mention a motion capture system but is it used at this point? I believe a high level testing approach description would be beneficial to describe your outcomes.